

# Functional characteristics, intercellular interactions and pathophysiological associations of astrocytes in Parkinson's disease

Ziying Li[1], Mengran Cao[1], Zhaoyang Yin[1], Xiaolei Li[2], Qinglu Wang[1], Panpan Dong[3] and Caixia Zhou[1]

[1] College of Sport and Health, Shandong Sport University, Jinan, Shandong, China
[2] College of Public Education, Shandong Medicine Technician College, Taian, China
[3] College of Basal Medical, Qilu Medical University, Zibo, Shandong, China

Corresponding authors
Panpan Dong,
18364346430@163.com
Caixia Zhou,
zhoucaixia@sdpei.edu.cn

## ABSTRACT

Parkinson's disease (PD) is the second most prevalent neurodegenerative disorder, and its incidence rate is increasing at an alarming rate. Astrocytes exhibit a complex array of functions that play a critical role in the pathogenesis and progression of PD. These multifaceted functions substantially influence disease development and exacerbation. Although numerous studies have investigated the role of astrocytes in PD, the interactions between astrocytes and other cellular or molecular factors during the course of neurological deterioration in PD have not been comprehensively characterized. Therefore, this review aims to highlight the roles and functional characteristics of astrocytes in PD. Astrocytes are involved in maintaining the blood-brain barrier, clearing α-synuclein, metabolizing glutamate and fatty acids, and protecting neurons. The interactions among astrocytes, microglia, and oligodendrocytes exert dual effects on PD progression. Additionally, the recently recognized phenomena of ferroptosis and cuproptosis have been associated with astrocytic activity. The interplay and regulatory mechanisms linking these forms of cell death to apoptosis and pyroptosis of immune cells in the brain warrant further investigation. This review is intended for researchers, healthcare professionals, and clinicians involved in the study and treatment of PD and its related complications. To ensure comprehensive and unbiased coverage, a systematic literature search was conducted using major scientific databases such as PubMed, Scopus, and Web of Science. Keywords included "Parkinson's disease", "astrocyte", "brain", "signaling mechanisms", and "α-synuclein". Articles were selected based on their relevance to astrocyte–PD interactions, while studies lacking scientific rigor or relevance were excluded. In summary, this review synthesizes current understanding of astrocytic function and mechanisms in PD and proposes potential therapeutic directions based on these insights.

## INTRODUCTION

Parkinson's disease (PD) predominantly affects middle-aged and elderly individuals and is characterized by movement-related disorders. These typically manifest as resting tremors, bradykinesia, and impaired postural balance. The underlying cause of these motor symptoms is the degeneration of neurons in specific brain regions, results in reduced dopamine production (*Wang et al., 2023*). Dopamine is a critical neurotransmitter.

Astrocytes are the most abundant glial cells in the brain. An increasing body of research suggests that astrocytes carry out a wide array of essential functions. These include regulating ionic homeostasis and secreting neurotrophic factors, both of which are critical for maintaining the physiological environment of the brain and supporting neuronal health (*Pehar et al., 2004*). Furthermore, certain genetic mutations associated with PD are believed to impair astrocyte function (*Booth, Hirst & Wade-Martins, 2017*). This review aims to examine the role of astrocytes in PD, focusing on the mechanisms underlying their dysfunction and the resulting pathological consequences (Fig. 1).

## SEARCH METHODOLOGY

Several key steps were undertaken to ensure that the literature review was both comprehensive and unbiased (Fig. 2). First, an extensive search was conducted across multiple academic databases (including PubMed, Scopus and Web of Science) to identify relevant trials, guidelines and research reports. The search terms included "Parkinson's disease," "astrocyte," "brain," "signaling mechanisms," and "α-synuclein" to capture a broad range of relevant sources. Literature that did not meet the inclusion criteria was excluded, including non-peer-reviewed publications, studies unrelated to astrocytes, non-English articles, and those focused primarily on Alzheimer's disease. Following this initial screening, selected articles were critically evaluated for their relevance and methodological quality. To minimize selection bias, studies from various disciplines and countries were included, encompassing both theoretical and applied perspectives. Articles were chosen based on their focus on the interaction between PD and astrocytes, while those with irrelevant content or insufficient experimental rigor were excluded. Efforts were also made to incorporate both foundational and recent publications to ensure a balanced and comprehensive overview. Additionally, a snowballing technique was employed by reviewing the reference lists of identified articles to locate further relevant publications that may have been missed during the initial database search, thereby enhancing the overall depth and breadth of the review.

## ASTROCYTES AND THE BLOOD-BRAIN BARRIER

The blood-brain barrier (BBB) is a critical interface between the cerebral parenchyma and the systemic circulation. It is regulated by a complex network comprising astrocytes, endothelial cells, and specialized pericytes, which together constitute the neurovascular units. This highly selective barrier tightly governs molecular trafficking, allowing the controlled influx of essential substances required for neural homeostasis while effectively restricting the entry of potentially harmful substances, thereby safeguarding neural tissue from toxic insults (*Iadecola, 2017*). Accumulating evidence has shown that astrocytes play

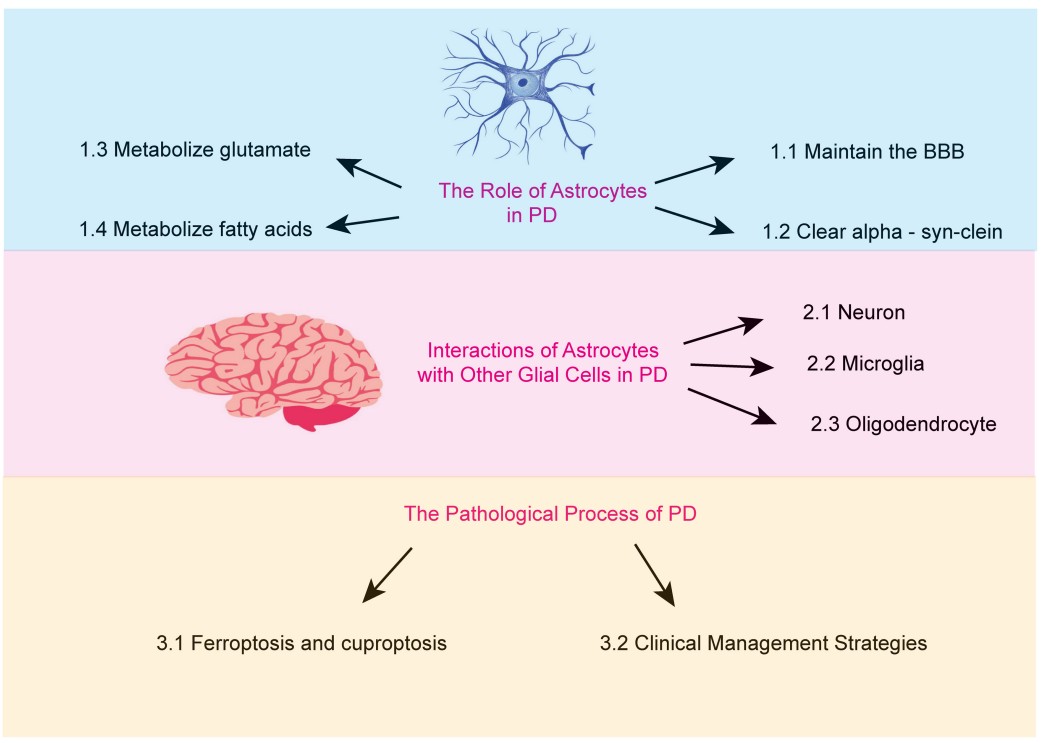

**Figure 1 The major topics and subtopics covered in our review.** (1) the functions of astrocytes, (2) the interactions between astrocytes and other glial cells in Parkinson's disease, and (3) the pathophysiological processes of Parkinson's disease.

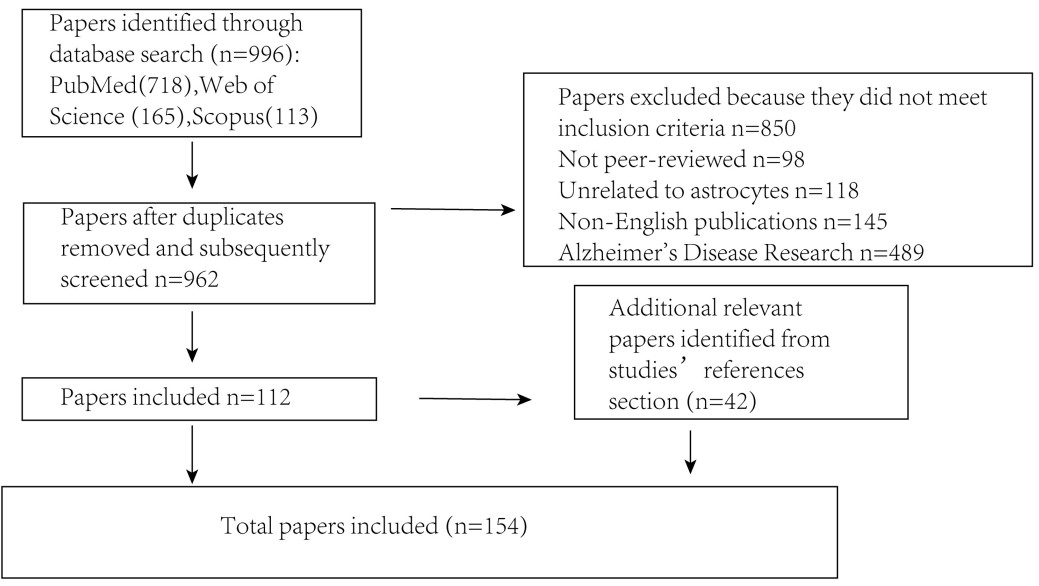

**Figure 2 Illustrates the procedure for sifting through articles following a database search.** The number of articles incorporated into or removed from the review process is presented at each stage.

a pivotal role in enhancing the functional integrity of the BBB (Fig. 1). Notably, in co-culture systems with endothelial cells, astrocytes have been demonstrated to induce the transdifferentiation of non-neurovascular endothelial cells into a brain-specific phenotype, thereby giving them BBB-like characteristics (Hayashi et al., 1997).

Apolipoprotein E (APOE), which is endogenously synthesized by astrocytes, is a multifunctional secreted macromolecule with a plethora of functions integral to maintaining cerebrovascular homeostasis. Under physiological conditions, APOE plays a critical role in preserving, the structural integrity of the BBB, primarily by modulating intercellular and extracellular interactions within the neurovascular unit. It is hypothesized that APOE proteins contribute to the stability of tight junctions in BBB endothelial cells. They achieve this by precisely regulating the complex metabolic processes associated with extracellular matrix components. Specifically, APOE has been implicated in the regulation of key matrix proteins such as collagen and laminin, which are essential for maintaining tight junction integrity. In humans, the APOE4 allele has been associated with increased BBB permeability (Jackson et al., 2022). Experimental studies using APOE4-transgenic mice have demonstrated that this allele induces tight junction disruption, abnormal overexpression of matrix metalloproteinase 9 (MMP9), and reduced vascular coverage of terminal peduncles. Collectively, these changes result in elevated BBB permeability, allowing peripheral inflammatory cytokines and immune cells to infiltrate the brain parenchyma, thereby exacerbating neuroinflammation and neuronal injury. Notably, astrocyte-specific deletion of APOE4 has been shown to reverse these pathological phenotypes (Jackson et al., 2022). These findings are particularly relevant for understanding the vascular dysfunctions associated with APOE4 in the context of PD, offering novel insights into the disease's underlying mechanisms. Further studies by Cooper et al. (2021) demonstrated that APOE4 has the highest binding affinity for low-density lipoprotein receptor-related protein 1 (LRP1), and that its lipidation state in the vascular microenvironment may influence its activity. Interestingly, the detrimental effects of astrocyte-derived APOE4 on BBB integrity may also occur via an LRP1-independent pathway, potentially involving alternative signaling mechanisms that activate MMP9.

Astrocytes can precisely regulate the functional activity of tight junction (TJ) proteins within the cytoplasmic environment of endothelial cells by secreting a diverse array of growth factors (Pivoriunas & Verkhratsky, 2021). The interaction between endothelial cells and astrocytes is therefore essential for maintaining the structural and functional integrity of the BBB. TJ proteins are critical for preventing transcellular diffusion of molecules through intercellular spaces and play a key role in defining the apical and basolateral domains of the plasma membrane, thereby supporting the establishment and maintenance of endothelial cell polarity (Wolburg & Lippoldt, 2002). Proper regulation of BBB permeability is vital for neuronal survival, as the barrier's selective properties effectively prevent harmful substances from entering the brain, thus preserving the neural microenvironment. Astrocytes contribute to this regulatory network by releasing cytokines and signaling molecules that finely modulate TJ protein expression, ensuring the continued structural and functional integrity of the BBB. Additionally, astrocytes are
involved in maintaining water and potassium ion homeostasis. Their perivascular end-feet extensively cover capillary surfaces and fulfill a variety of essential functions linked to BBB physiology. Notably, these end-feet express aquaporin (AQP) channels (*Lanciotti et al., 2013*), In humans, aquaporins are classified as AQP1 and AQP4, with AQP4 being predominantly expressed in the perivascular astrocytic end-feet (*Lanciotti et al., 2013*). Reduced AQP4 expression has been associated with disrupted potassium ($K^+$) homeostasis, indicating its key role in maintaining ionic balance (*Binder et al., 2006*). Furthermore, astrocytes secrete antioxidants, particularly glutathion, which are thought to influence BBB permeability. Glutathione protects neurons from neurotoxic damage by reducing the ability of harmful molecules to penetrate the brain (*Agarwal & Shukla, 1999*).

In the cerebral parenchyma of patients with PD, the integrity of BBB is compromised. Subtle disruptions have been identified in brain regions critical to PD pathophysiology, including the substantia nigra, white matter tracts, and posterior cortical areas (*Aflaki et al., 2020*). These findings suggest a complex association between BBB breakdown and PD development. As previously discussed, astrocyte dysfunction is a key contributor to BBB disruption. In the healthy cerebral cortex, astrocytes promote the secretion of TJ proteins through the release of cytokines. However, under the pathological conditions of PD, reactive astrocytes exhibit significantly reduced expression of growth factors (*Pehar et al., 2004*). This decline in growth factor expression leads to a corresponding reduction in TJ protein production. The resultant TJ protein deficiency increases BBB permeability, rendering the brain more vulnerable to environmental toxins and harmful substances. These external insults can initiate a cascade of deleterious events that ultimately lead to neuronal death.

Conventional therapeutic approaches for PD, such as deep-brain electrical stimulation and dopamine replacement therapy, provide symptomatic relief but do not offer a cure (*Khor et al., 2024*). Recently, bioengineered exosomes have emerged as a promising therapeutic strategy due to their ability to cross the BBB. Their innate permeability, coupled with excellent biocompatibility, makes them ideal carriers for delivering therapeutic agents. Consequently, bioengineered exosomes hold considerable potential for the effective treatment of PD, representing a novel and potentially more efficient approach to managing this neurodegenerative disorder (*Afzal et al., 2025*).

## ASTROCYTES AND ALPHA-SYNUCLEIN

Alpha-Synuclein (α-syn) is a highly dynamic intracellular protein that normally exists in monomeric or multimeric forms (*Afzal et al., 2025*; *Dettmer, 2018*). However, when aggregated, α-syn becomes neurotoxic (*Stefanis, 2012*). Extracellular α-syn can be internalized by neighboring neurons, potentially propagating pathology (*Xie et al., 2022*). Astrocytes play a protective role by attenuating neuronal α-syn pathology through uptake and regulation of protein homeostasis mechanisms associated with the formation, trafficking, depolymerization, and clearance of toxic α-syn aggregates, as demonstrated in various *in vitro* and *in vivo* models of α-synucleinopathies (*Yang et al., 2022*). Astrocytes also reduce neurotoxicity by phagocytosing extracellular α-syn (*Hua et al., 2019*; *Yang et al., 2022*), and secrete molecules capable of degrading extracellular α-syn. Notably,

protein disulfide isomerase, a thiol-disulfide oxidoreductase highly expressed in astrocytes, has been shown to inhibit α-syn fibrillation (*Serrano et al., 2020*). However, excessive α-syn can induce astrocytic activation and transformation into a pro-inflammatory phenotype, which negatively affects neuronal survival (*Chou et al., 2021*). In PD, α-syn misfolds and assembles into oligomers and fibrils that accumulate within neurons and glial cells, eventually forming Lewy bodies—a pathological hallmark of the disease. These aggregates activate astrocytes, causing significant morphological and functional changes. Activated astrocytes release inflammatory cytokines and oxidative stress-related molecules, which, through a feedback mechanism, promote further α-syn misfolding and aggregation. This establishes a self-amplifying cycle (*da Fonseca, Villar-Pique & Outeiro, 2015*; *Parmar, 2022*), that intensifies neuronal damage, accelerates PD progression, and worsens clinical outcomes (*Yan et al., 2024*).

Given this duality, reactive astrocytes cannot be regarded solely as protective; under certain conditions, they may exert neurodegenerative effects. Therefore, therapeutic strategies involving astrocytes must carefully consider this complexity (*Wang, Sun & Dettmer, 2023*). Additionally, intracellular pathological α-syn can be secreted into the extracellular environment, activating neighboring neurons and glial cells (*He et al., 2024*). This creates a microenvironment persistently influenced by α-syn, contributing to chronic neuroinflammation and neurodegeneration. Thus, α-syn functions as a critical link between inflammatory and degenerative processes in PD (*Serrano et al., 2020*).

## GLUTAMATE METABOLISM IN ASTROCYTES

Glutamate is a major excitatory neurotransmitter in the brain (Fig. 3). However, excessive glutamate accumulation in the synaptic cleft is neurotoxic due to overstimulation of N-methyl-D-aspartate (NMDA) receptors, leading to elevated intracellular calcium, activation of cell death signaling pathways, and ultimately, neuronal necrosis and apoptosis (*Shen et al., 2022*). Glutamate excitotoxicity is a key contributor to dopaminergic neuron degeneration in PD. In PD, dysfunction or loss of glutamate dehydrogenase (GDH)—a critical enzyme in glutamate metabolism—has been observed. GDH plays a central role in bridging the GABA–glutamine cycle with the tricarboxylic acid (TCA) cycle. As the brain is the most energy-demanding organ, and neurons rely heavily on oxidative metabolism, GDH is essential for sustaining energy production, especially under conditions of increased demand. It facilitates glutamate metabolism and regulates mitochondrial respiratory activity, thus supporting neuronal viability and neuroplasticity (*Michaelis et al., 2011*).

Astrocytes play a crucial role in maintaining the brain's chemical balance by efficiently removing excess glutamate from the extracellular space. This function is primarily mediated by excitatory amino acid transporters (EAATs) located on the astrocytic membrane. By regulating extracellular glutamate levels, astrocytes prevent glutamate excitotoxicity—a pathological condition in which excessive glutamate overstimulates receptors, leading to neuronal damage and death. This regulatory mechanism is essential for preserving neuronal integrity and ensuring that glutamate concentrations remain

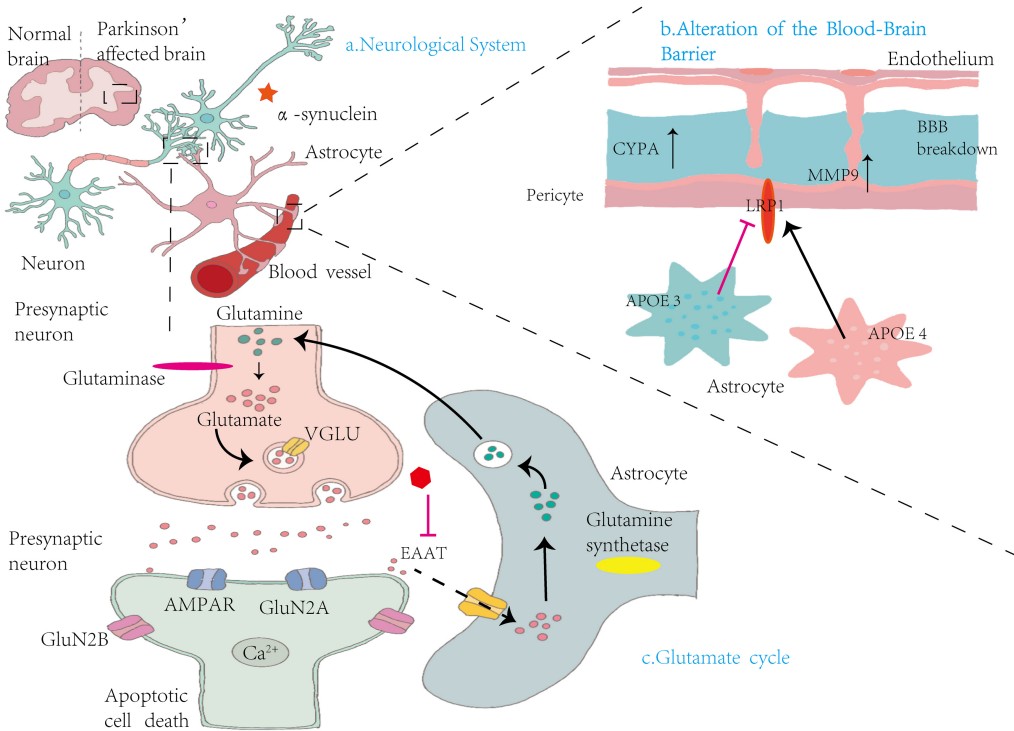

**Figure 3 Mechanisms by which astrocytes influence nervous system alterations, blood–brain barrier integrity, and glutamate cycling in Parkinson's disease.** (A) Neurological System: This panel compares the brains of healthy individuals with those affected by PD. In PD, there is notable accumulation of α-syn in the vicinity of neurons, contributing to neurodegeneration. (B) Alterations in the Blood-Brain Barrier: This section illustrates the structural components of the BBB, including endothelial cells, pericytes, and astrocytes. Elevated levels of cytochrome P450 (CYP4A) in pericytes lead to increased expression of MMP9, resulting in BBB disruption. Concurrently, astrocytic expression of apolipoprotein E isoforms—APOE3 and APOE4—demonstrates functional differences. The LRP1 modulates APOE function, influencing BBB stability and permeability. (C) Glutamate cycle: This panel illustrates the glutamate–glutamine cycle among presynaptic neurons, synaptic clefts, and astrocytes. Within presynaptic neurons, glutaminase converts glutamine into glutamate, which is then packaged into synaptic vesicles by vesicular glutamate transporters (VGLUT). Upon release, glutamate activates AMPA receptors (AMPARs) and NMDA receptor subunits (GluN2A and GluN2B) on postsynaptic neurons, facilitating $Ca^{2+}$ influx and potentially inducing apoptosis. Astrocytes absorb excess glutamate *via* excitatory amino acid transporters (EAATs) and convert it into glutamine through glutamine synthetase. The resulting glutamine is shuttled back to presynaptic neurons, completing the cycle. This process maintains synaptic homeostasis and prevents excitotoxicity.

within a physiological range, thereby supporting normal synaptic transmission and neural function (*Cuellar-Santoyo et al., 2022*).

To further investigate the neuroprotective role of astrocytes in mitigating glutamate toxicity, *Rosenberg & Aizenman (1989)* demonstrated that cortical neurons cultured in astrocyte-poor media were approximately 100-fold more susceptible to prolonged glutamate exposure than those cultured in astrocyte-rich conditions. This finding highlights the indispensable role of astrocytes in protecting neurons from excitotoxic damage and maintaining the physiological stability of the nervous system (*Rosenberg & Aizenman, 1989*). Astrocytes also participate in neurotransmitter recycling

and energy metabolism through dual metabolic pathways. Through the GABA–glutamine–glutamate (GGG) cycle, astrocytes convert absorbed glutamate or GABA into glutamine, which is subsequently shuttled back to neurons. Alternatively, these neurotransmitters can be metabolized into α-ketoglutarate (α-KG) *via* the TCA cycle, contributing to ATP synthesis. This dual role highlights the capacity of astrocytes to regulate extracellular neurotransmitter levels while simultaneously supporting the brain's high energy demands (*Zhou & Danbolt, 2014*). The GGG cycle maintains a dynamic balance of glutamate, glutamine, and GABA between astrocytes and neurons. Specifically, glutamate and GABA are transported from neurons to astrocytes, while glutamine is returned to neurons, ensuring proper synaptic communication and neural function (*Walls et al., 2015*). Disruption of the GGG cycle has been implicated in various pathological conditions, including neurodevelopmental and neurodegenerative disorders such as PD and Alzheimer's disease, as well as psychiatric and cognitive impairments. Given its central role in neurotransmitter regulation, GGG cycle dysregulation can have widespread consequences for brain function and systemic health (*Lander et al., 2020*; *Rappeneau et al., 2016*). Emerging evidence suggests that glutamatergic astrocytes, despite being relatively rare, play a pivotal role in modulating the integrity of cortico-hippocampal and nigrostriatal circuits under both physiological and pathological conditions. In PD, the subthalamic nucleus, a key node in the basal ganglia–thalamo–cortical loop, exhibits pathological hyperactivation that is closely linked to glutamatergic astrocyte dysfunction. This connection highlights a critical role for glutamatergic astrocytes in maintaining circuit homeostasis and positions them as promising therapeutic targets for PD. Harnessing the unique functional properties of these astrocytes may enable the development of novel interventions aimed at restoring disrupted neural connectivity and alleviating both motor and non-motor symptoms in PD (*de Ceglia et al., 2023*). Beyond glutamate regulation, astrocytes contribute to multiple aspects of brain metabolism, underscoring their broad influence on neural health and signaling. This includes key roles in fatty acid metabolism, which is integral to sustaining the physiological function of the nervous system.

## Astrocytes and fatty acid metabolism

In the brain, excess fatty acids are stored in lipid droplets (LDs) as neutral lipids such as triglycerides to prevent cytotoxic accumulation (*Ralhan et al., 2021*). These LDs facilitate the transport of fatty acids into mitochondria for energy production (*Schonfeld & Reiser, 2013*), thereby preventing excessive fatty acid accumulation and inhibiting lipid peroxidation—an oxidative process that generates toxic lipid peroxides (*Bailey et al., 2015*). Notably, neurons exhibit extremely low levels of LDs (*Ioannou et al., 2019*), and possess limited capacity to metabolize fatty acids, as the resulting oxidative by-products are particularly harmful to neuronal cells (*Ding et al., 2024*). Consequently, neurons rely on alternative mechanisms to offload excess fatty acids. In the brains of PD patients, fatty acids are especially prone to lipid peroxidation due to heightened oxidative stress. Reactive oxygen species (ROS) attack the unsaturated double bonds in fatty acids, initiating a chain reaction that leads to the formation of toxic peroxidation products such as

malondialdehyde (MDA). These lipid peroxides compromise the integrity of cellular membranes and organelles, disrupt cellular homeostasis, and ultimately trigger apoptosis. This oxidative stress-induced lipid peroxidation is considered a key pathological mechanism underlying neurodegeneration in PD, making it a potential therapeutic target.

Studies have shown that fatty acids can be transferred from neurons to astrocytes, which are better equipped to handle oxidative stress (*Belanger & Magistretti, 2009*). Astrocytes express high levels of genes related to β-oxidation, a critical pathway for fatty acid catabolism. Although the underlying mechanisms are not yet fully elucidated, it is known that lipid transporter proteins such as APOE facilitate the transfer of fatty acids from overactive neurons to astrocytes. This astrocyte-mediated metabolic process helps protect neurons from the excitotoxic effects of lipid accumulation (*Ioannou et al., 2019*). Moreover, astrocytes express NMDA receptors—part of the glutamate receptor family. Ioannou and colleagues observed that glutamate exposure reduces LD content in astrocytes, suggesting increased fatty acid utilization and concurrent ATP generation. Glutamate may also stimulate ATP release into the synaptic cleft, influencing neurotransmission through receptor interactions. For example, ATP can activate P2X receptors, suppress dopaminergic synaptic activity, and ultimately inhibit NMDA receptor function (*Zhang et al., 2023*; *Koizumi et al., 2003*; *Lalo et al., 2016*). This cascade illustrates a complex interplay between glutamate signaling, lipid metabolism, and energy regulation within the synaptic microenvironment. The astrocytic conversion of fatty acids into ATP may thus represent a protective mechanism to counter excitotoxic stress in hyperactive neurons. This process is critical for maintaining neuronal function and preventing the metabolic and excitotoxic damage characteristic of neurodegenerative disorders such as PD.

## Astrocytes and neurons

Astrocytes play a pivotal role in promoting neuronal survival through multiple intricate mechanisms. These include the secretion of neurotrophic factors and antioxidants, that support neuronal growth, differentiation, and protection against oxidative stress, as demonstrated in numerous studies (*Mederos, Gonzalez-Arias & Perea, 2018*). Astrocytes also facilitate the clearance of α-syn—a protein whose pathological aggregation is implicated in several neurodegenerative disorders-and are actively involved in regulating glutamate levels of this neurotransmitter to prevent excitotoxicity, as well as in fatty acid metabolism, which may contribute to energy homeostasis and protection against excitotoxic damage in neurons (*Prunell & Olivera-Bravo, 2022*). Additionally, astrocytes are capable of transferring healthy mitochondria to neurons, thereby enhancing neuronal energy production and resilience. However, astrocyte function is not homogeneous across the brain; their roles vary significantly depending on their anatomical localization. Regional heterogeneity in astrocyte phenotype reflects the distinct physiological and pathological demands of different brain regions, underscoring the complexity of astrocyte–neuron interactions and their significance in both normal function and disease pathophysiology (*Kostuk, Cai & Iacovitti, 2019*). For example, reactive astrocytes—those undergoing structural and functional changes in response to injury or disease—exhibit alterations in gene expression, protein profiles, morphology, and physiological activity.

These changes, part of the brain's intrinsic defense mechanisms, are aimed at protecting neurons and restoring tissue homeostasis (*Sonninen et al., 2020*). However, in certain contexts, reactive astrocytes can contribute to disease progression rather than resolution, highlighting their dual nature in neurodegenerative conditions (*Chiareli et al., 2021*). In this review, reactive astrocytes are defined as those emerging under pathological conditions. Upon activation, they secrete various neurotrophic factors (NTFs) that promote neuronal survival, growth, and differentiation—responses critical for neuronal protection in damaged microenvironments. Beyond NTF release, reactive astrocytes also participate in extracellular matrix remodeling and immune regulation, both of which influence the overall pathophysiological response to injury (*Zhang et al., 2023*). Among the secreted NTFs, glial cell line-derived neurotrophic factor (GDNF) has been the most extensively studied. GDNF exhibits potent neuroprotective and regenerative effects, particularly on dopaminergic neurons, which are highly vulnerable in disorders such as PD. Detailed investigations have confirmed GDNF's therapeutic potential and its involvement in mechanisms underlying neuronal regeneration (*Deierborg et al., 2008*). A foundational study by *Lin et al. (1994)* showed that GDNF enhances dopamine reuptake by dopaminergic neurons at the synapse, thereby improving dopamine utilization and supporting neuronal viability—an essential function in the context of PD (*Airaksinen & Saarma, 2002*; *Lin et al., 1994*). Other NTFs, such as midbrain astrocyte-derived neurotrophic factor (MANF) and basic fibroblast growth factor, also contribute to neuroprotection. MANF, predominantly expressed in the midbrain, supports specific neuronal populations (*Petrova et al., 2004*), while basic fibroblast growth factor participates in cell proliferation, differentiation, and survival. Although their protective effects may not be as extensively documented as GDNF, these factors work synergistically or independently within the broader neuroprotective network, playing an essential role in maintaining the structural and functional integrity of the nervous system (*Grothe & Timmer, 2007*).

Astrocytes play a critical role in protecting the brain against oxidative stress. Given the brain's high metabolic activity, approximately 1–2% of mitochondrial oxygen is converted into ROS rather than water. This continuous ROS production can cause oxidative damage to cellular components such as lipids, proteins, and DNA. Astrocytes are equipped with robust antioxidant defense systems, including the production of key enzymes such as superoxide dismutase, catalase, and glutathione peroxidase. These enzymes work in concert to neutralize ROS, thereby safeguarding neurons from oxidative damage. Their capacity to buffer ROS is vital for preserving normal brain function and preventing the onset and progression of neurodegenerative diseases linked to oxidative stress (*Richter, 1992*). An imbalance between oxidant and antioxidant systems, called oxidative stress, is closely associated with the pathogenesis of PD. In PD, oxidative stress results in the oxidation of lipids, proteins, and nucleic acids within dopaminergic neurons, impairing essential processes such as energy metabolism, protein folding, and neurotransmitter regulation. Over time, these cumulative effects contribute to the progressive degeneration of dopaminergic neurons, which is a hallmark of PD (*Barnham, Masters & Bush, 2004*). Understanding this relationship between the oxidant-antioxidant imbalance and PD

progression is crucial for developing therapeutic strategies aimed at halting or reversing the disease process.

Ferroptosis is associated with astrocyte dysfunction. As a component of the blood-brain barrier, astrocytes can mediate iron influx into the brain and regulate the transport of various types of iron. However, in PD, dysfunction of astrocytes and iron transporters may lead to selective iron deposition in the substantia nigra pars compacta (SNPC). Subsequently, the inherent redox-active chemical property of iron lays the foundation for its potential toxicity. Accumulated iron drives the expansion of mitochondrial ROS and lipid peroxidation through the Fenton reaction, ultimately resulting in damage to DNA, lipids, and proteins (*Zhang et al., 2025*).

Studies have shown that astrocytes exert a neuroprotective effect on neuromelanin-containing dopaminergic neurons in PD. Specifically, during the synthesis of neuromelanin, the oxidation of dopamine to form adrenochrome plays a significant role in the degeneration of neuromelanin-containing dopaminergic neurons within the nigrostriatal system. It has been proposed that adrenochrome, as an endogenous neurotoxin, can initiate the neurodegenerative process in idiopathic PD by triggering a series of pathological reactions, including mitochondrial dysfunction, oxidative stress, neuroinflammation, impaired function of the lysosomal and proteasomal protein degradation systems, endoplasmic reticulum stress, and the formation of neurotoxic α-synuclein oligomers (*Huenchuguala & Segura-Aguilar, 2024a*; *Wang et al., 2022a*). Therefore, preventing the oxidation of dopamine to adrenochrome may represent a key neuroprotective strategy for safeguarding the aforementioned neurons that are lost in the substantia nigra in PD (*Huenchuguala & Segura-Aguilar, 2024b*). Astrocytes play an important role in protecting neurons against oxidative stress damage, and one of their mechanisms involves the secretion of glutathione precursors (glutathione being a crucial antioxidant) (*Cuevas et al., 2015*; *Valdes et al., 2021*). Meanwhile, neuromelanin-containing dopaminergic neurons are also affected by the neurotoxicity of adrenochrome upon exposure, which in turn leads to oxidative stress, mitochondrial dysfunction, the formation of neurotoxic α-synuclein oligomers, impaired function of the lysosomal and proteasomal protein degradation systems, neuroinflammation, and endoplasmic reticulum stress (*Capucciati et al., 2021*). However, astrocytes can exert a further protective effect by secreting exosomes loaded with glutathione S-transferase M2-2 (*Segura-Aguilar et al., 2022*); these exosomes, together with DT-diaphorase, can penetrate dopaminergic neurons, thereby preventing the neurotoxic effects of adrenochrome (*Segura-Aguilar & Mannervik, 2023*).

Among the various mechanisms for ROS detoxification, glutathione production is of particular importance. Astrocytes supply extracellular glutathione to neurons, serving as a critical line of defense against oxidative injury. Glutathione, a tripeptide composed of cysteine, glycine, and glutamate, acts as a potent antioxidant by directly scavenging ROS and participating in glutathione-dependent enzymatic reactions, notably those catalyzed by glutathione peroxidase. This astrocyte-to-neuron transfer of glutathione constitutes an

essential intercellular mechanism for protecting neuronal structures from oxidative stress (*Wang, Sun & Dettmer, 2023*). In oxidative environments, astrocytes can also export glutathione disulfide—the oxidized form of glutathione—which further contributes to ROS neutralization. When released into the extracellular space, glutathione disulfide interacts with ROS, reducing their reactivity and preventing damage to neuronal membranes, proteins, and DNA. This export mechanism supports redox balance within the neuronal microenvironment and plays a significant role in mitigating oxidative stress-related neurotoxicity (*Dringen & Hirrlinger, 2003*).

Another potential neuroprotective pathway involving astrocytes is the endogenous cannabinoid (eCB) system (*Schuele et al., 2022*). Endocannabinoids are lipid-derived signaling molecules that play key roles in various physiological and pathological processes, including inflammation, neurotransmission, and neuroprotection (*Kano et al., 2009*). The eCB system comprises endogenous cannabinoids, their receptors—such as cannabinoid receptors 1 and 2 (CB1R and CB2R)—and the enzymes responsible for their synthesis and degradation. In response to injury, inflammation, or neurodegeneration, this system is activated, and endocannabinoids bind to their receptors, initiating intracellular cascades that suppress pro-inflammatory cytokine production, reduce oxidative stress, and modulate synaptic activity. These combined actions promote neuronal survival and recovery, making the eCB system a compelling therapeutic target for neurological disorders (*Eljaschewitsch et al., 2006*; *Panikashvili et al., 2001*). One of the primary endocannabinoids, 2-arachidonoylglycerol (2-AG), exerts its neuroprotective effects *via* the CB1 receptor. Predominantly expressed on neurons, CB1R activation by 2-AG regulates ion channel activity, inhibits neurotransmitter release, and attenuates ROS and inflammatory cytokine production (*Chen, Zhang & Chen, 2011*). Astrocytes play a vital role in this process. The gene diacylglycerol lipase alpha (DAGLA), which encodes a key enzyme for 2-AG biosynthesis, is expressed in astrocytes. Deletion of DAGLA in astrocytes impairs neuronal survival and neurogenesis, suggesting that astrocyte-derived 2-AG is necessary for supporting neural health. DAGLA-catalyzed production of 2-AG in astrocytes appears to be essential for both preserving existing neurons and promoting the generation of new ones. Its disruption leads to adverse outcomes in neuronal populations, highlighting the significance of astrocyte-mediated eCB signaling in maintaining a healthy neural environment. This connection between astrocytes, DAGLA-dependent 2-AG synthesis, and neurogenesis positions the endocannabinoid pathway as a promising target for novel therapeutic strategies aimed at treating neurodegenerative conditions or disorders associated with impaired neuronal regeneration (*Schuele et al., 2022*).

## ASTROCYTES AND MICROGLIA: ROLE OF CELLS IN PD

### Proinflammatory phenotype and inflammatory response of microglia

A study conducted by *Mogi et al. (1996)* reported elevated levels of inflammatory cytokines, including tumor necrosis factor-alpha (TNF-$\alpha$), interleukin-1 beta (IL-1$\beta$), and interleukin-6 (IL-6), in both the brain tissue and cerebrospinal fluid of patients with PD.

These cytokines are key mediators of the inflammatory response, and their increased expression in PD indicates an ongoing neuroinflammatory process. Within dopaminergic neurons, elevated TNF-α levels can initiate inflammatory cascades, promoting oxidative stress and activating apoptotic pathways. As these neurons progressively degenerate due to TNF-α exposure, the expression of tyrosine hydroxylase—a rate-limiting enzyme in dopamine synthesis—also declines. This reduction serves as a biomarker for dopaminergic neuronal damage, further underscoring TNF-α's role in neurodegenerative processes affecting the dopaminergic system (*Daubner, Le & Wang, 2011*). In PD, the underlying principle of the observations is that a decrease in tyrosine hydroxylase corresponds to a decline in dopamine levels. Tyrosine hydroxylase is the rate—limiting enzyme in dopamine biosynthesis. In PD, the progressive loss of dopaminergic neurons leads to a reduction in the amount of tyrosine hydroxylase available. As a result, the conversion of tyrosine to L-DOPA, the precursor of dopamine, is impeded. This directly causes a decrease in dopamine production, which is a characteristic feature of PD. The correlation between reduced tyrosine hydroxylase and diminished dopamine levels is fundamental to understanding the pathophysiology of PD and is also crucial for developing diagnostic methods and therapeutic strategies aimed at restoring dopamine function in the brain. Astrocytes, in addition to being immunoreactive cells, play a crucial role in regulating brain inflammation. They are involved in the production and release of inflammatory cytokines, while microglia are generally regarded as the main resident immune cells in the brain. In PD, the loss of tyrosine hydroxylase directly correlates with reduced dopamine production, as the enzyme is essential for converting tyrosine to L-DOPA, the precursor of dopamine. Understanding this relationship is critical for elucidating PD pathophysiology and for developing diagnostic and therapeutic strategies aimed at restoring dopamine function in the brain.

Astrocytes, although primarily known for their supportive functions, also play an active role in modulating neuroinflammation. Alongside microglia—the principal immune cells of the central nervous system—astrocytes contribute to the production and regulation of inflammatory cytokines. In PD, astrocytes appear to counteract excessive microglial activation through a negative feedback mechanism. Upon encountering pathological stimuli, microglia become activated and secrete large quantities of pro-inflammatory mediators such as TNF-α, IL-1β, IL-6, and ROS. These molecules disrupt normal neuronal function, induce apoptosis, and perpetuate further immune activation. Specifically, TNF-α can impair synaptic signaling, IL-1β can interfere with neurotransmitter regulation and synaptic plasticity, and ROS can directly damage neuronal membranes, proteins, and DNA.

Astrocytes, however, counteract this potentially harmful process. They sense the activation of microglia and the release of inflammatory mediators. Through a negative feedback loop, astrocytes secrete factors that can suppress microglial activation, thereby reducing the production of pro-inflammatory cytokines and ROS. This regulatory mechanism is crucial for maintaining the balance of the neural microenvironment and for

protecting neurons from the deleterious effects of excessive microglial activity in PD (*Block, Zecca & Hong, 2007*). Numerous *in vitro* studies have demonstrated that either astrocyte-conditioned medium or the direct presence of astrocytes can attenuate microglial responses to various pro-inflammatory stimuli. When microglia are exposed to agents such as lipopolysaccharide (LPS) or inflammatory cytokines, astrocyte-derived substances in the medium or physical interaction with astrocytes significantly mitigate the inflammatory response. Astrocytes achieve this by secreting a range of anti-inflammatory and immunomodulatory molecules, including transforming growth factor-beta (TGF-β), interleukin-10 (IL-10), and prostaglandin E2 (PGE2). TGF-β downregulates the expression of pro-inflammatory cytokines in microglia, IL-10 exerts potent anti-inflammatory effects by inhibiting immune-related signaling pathways, and PGE2 modulates microglial function by reducing the production of ROS and inflammatory mediators. This attenuation of microglial activation is essential for sustaining the neuroimmune equilibrium. In neurodegenerative diseases, persistent microglial activation can lead to chronic neuroinflammation, which exacerbates neuronal injury. Thus, the capacity of astrocytes to dampen microglial responses underscores their protective role in the central nervous system (*Vincent, Tilders & Van Dam, 1997*). However, in PD, reactive microglia are present in substantial numbers (*McGeer et al., 1988*). In many pathological contexts, these activated microglia release cytokines such as interleukin-1α (IL-1α), which can transform astrocytes into a pro-inflammatory phenotype. These reactive, neurotoxic astrocytes not only increase the production of pro-inflammatory cytokines but also cease releasing neurotrophic factors and antioxidants, thereby further aggravating neurotoxicity and neuronal degeneration (*Liddelow et al., 2017*).

Developing an inhibitor to prevent the transformation of astrocytes into a reactive, pro-inflammatory phenotype represents a plausible and promising therapeutic strategy. Preliminary studies have identified molecules capable of inhibiting this activation. For example, *Chung et al. (2017)* demonstrated in an MPTP-induced PD rat model that intraperitoneal administration of capsaicin reduced the expression of inflammatory cytokines such as IL-1β in microglia. This effect is mediated through the activation of transient receptor potential vanilloid 1 (TRPV1) receptor—commonly known as capsaicin receptors—located in the brain. These findings suggest that modulating TRPV1 activity may offer a viable approach to suppressing reactive astrocyte activation and managing neuroinflammation-related neurological disorders (*Chung et al., 2017*). In PD, microglia-derived cytokines are known to activate pro-inflammatory astrocytes. Therefore, a therapeutic approach aimed at reducing astrocyte activation, while maintaining the regulatory balance between NF-κB and nuclear factor erythroid 2–related factor 2 (Nrf2) signaling pathways within astrocytes, could be particularly effective. Nrf2 activation can attenuate the harmful effects of NF-κB by promoting the release of anti-inflammatory and antioxidant factors (*Park et al., 2021*). Furthermore, Nrf2 has been shown to suppress microglial hyperactivation, thereby reducing the inflammatory phenotype of astrocytes
(*McGeer et al., 1988*; *Yang, Yang & Zhang, 2022*). However, the activation of Nrf2 is dependent on DJ-1, a multifunctional protein frequently mutated in familial PD. Under normal conditions, DJ-1 facilitates Nrf2 activation by promoting its nuclear translocation, where it binds to antioxidant response elements (AREs) and upregulates the expression of cytoprotective and antioxidant genes. Mutations in DJ-1 impair this function, thereby hindering Nrf2 activation and reducing the cell's capacity to combat oxidative stress. Given that oxidative stress is a central factor in PD pathogenesis, especially in familial forms of the disease, disruptions in the DJ-1–Nrf2 signaling axis may significantly contribute to neurodegeneration. Thus, understanding and targeting this molecular relationship may open new avenues for therapeutic development aimed at restoring redox balance and controlling glial reactivity in PD (*Yang, Yang & Zhang, 2022*).

### Microglia and α-synuclein

The abnormal folding of α-syn initiates its pathological accumulation, which subsequently triggers a cascade of events that drive neuroinflammation and promote neurodegenerative processes, ultimately contributing to the onset and progression of PD. Microglia, as essential immune cells of the central nervous system, play a pivotal role in orchestrating neuroinflammatory responses and maintaining cerebral homeostasis by removing cellular debris and potentially harmful substances from the brain environment. Upon detecting misfolded and aggregated α-syn, microglia initiate a series of responses involving inflammatory activation, cellular proliferation, and the mobilization of clearance mechanisms. These processes—although intended to preserve tissue integrity—exert a profound influence on disease progression in PD, as chronic microglial activation and impaired clearance of α-syn can exacerbate neuronal dysfunction and degeneration.

However, exposure to α-syn aggregates also activates microglia, initiating a state of chronic neuroinflammation that further exacerbates neurodegeneration.

In the context of neurodegenerative diseases such as PD, the presence of aggregated α-syn is a critical pathological hallmark. Microglia, the resident immune cells of the central nervous system, recognize these abnormal α-syn aggregates as danger signals. Upon detection, microglia transition into an activated state, initiating a complex response that involves the upregulation of immune-related genes and the release of inflammatory mediators. This activation leads to chronic neuroinflammation, characterized by the sustained release of pro-inflammatory cytokines, chemokines, and ROS. These factors not only cause direct neuronal damage but also disrupt the neural microenvironment, thereby accelerating neurodegeneration and contributing to the progressive loss of neuronal function (*Thi Lai et al., 2024*). Microglia are also involved in the clearance of α-syn aggregates through phagocytosis and autophagy. Phagocytosis allows microglia to engulf extracellular aggregates, while autophagy enables intracellular degradation and recycling of misfolded proteins. However, in PD, these essential clearance mechanisms are often impaired. Genetic mutations linked to PD may compromise the normal function of proteins involved in these pathways, and the oxidative stress commonly observed in PD

can damage the cellular machinery required for effective aggregate removal. Consequently, α-syn accumulates in the brain, perpetuating a cycle of inflammation and neurodegeneration. This accumulation further amplifies the release of inflammatory mediators, worsening the disease pathology (*Badanjak et al., 2021*; *Lv et al., 2023*). Additionally, microglia may contribute to the propagation of α-syn pathology by facilitating the intercellular spread of aggregates across brain regions (*Zheng & Zhang, 2021*). Therefore, a comprehensive understanding of the balance between microglial activation, aggregate clearance, and pathological propagation is essential for designing therapeutic strategies aimed at modulating microglial function to mitigate disease progression in PD.

## Cellular pyroptosis and PD in microglia and astrocytes

The development of PD can be influenced by both environmental and genetic factors (*Goldman, 2014*; *Wang et al., 2019*), with neuronal loss being a key pathological hallmark. Accumulating evidence suggests that programmed cell death (PCD) is a primary mechanism underlying this loss (*Moujalled, Strasser & Liddell, 2021*). Among the forms of PCD, cellular pyroptosis—a highly inflammatory mode of cell death—has been shown to play a pivotal role in the degeneration of dopaminergic neurons in PD. Pyroptosis is triggered by the activation of inflammasomes and inflammatory vesicles (*Han et al., 2023*; *Wang et al., 2019*; *Wu et al., 2023*), and is closely associated with the release of pro-inflammatory factors and the activation of glial cells, which collectively contribute to dopaminergic neuronal death (*Hu et al., 2022*). A defining feature of pyroptosis is the cytolytic action of gasdermin D (GSDMD), which forms membrane pores leading to cell rupture (*Kesavardhana, Malireddi & Kanneganti, 2020*). GSDMD has been implicated in various neurological disorders, particularly by influencing microglial activation. Studies on GSDMD-mediated pyroptosis in PD have primarily focused on microglia (*Anderson et al., 2021*; *Feng et al., 2021*; *Gordon et al., 2018*). Damaged dopaminergic neurons release molecules such as matrix metalloproteinase 3 (MMP3), α-syn, neuromelanin, and ATP, all of which can activate microglia (*Block, Zecca & Hong, 2007*; *Glass et al., 2010*). Microglia are highly dynamic and exhibit substantial phenotypic plasticity, with their functional state varying based on disease stage and severity. Under normal physiological conditions, microglia display a ramified morphology and perform immune surveillance in a quiescent state. Upon encountering pathological stimuli, such as those present in PD, they become activated. While early activation may serve a protective role by clearing cellular debris, prolonged or excessive activation leads to a shift in phenotype, marked by altered expression of core markers, including the upregulation of inflammatory genes and downregulation of homeostatic functions, which plays a critical role in disease progression and therapeutic targeting (*Paolicelli et al., 2022*). Similarly, astrocytes exhibit heterogeneity under both physiological and pathological conditions. Activated microglia release cytokines such as IL-1α, TNF-α, and C1q, which are both necessary and sufficient to induce the formation of reactive astrocytes. Once activated, these astrocytes lose their

neuroprotective functions, including support for neuronal survival, synaptogenesis, and phagocytosis, and instead acquire neurotoxic properties. For instance, lipopolysaccharide (LPS)-stimulated microglia can induce astrocytes to release soluble toxic factors that directly promote neuronal and oligodendrocyte death. This microglia–astrocyte crosstalk and the resulting transformation of astrocytic function have significant implications for the progression of neurodegenerative diseases such as PD. Emerging evidence also points to the regulatory role of long non-coding RNAs (lncRNAs) in PD pathogenesis. LncRNAs have been shown to modulate the expression of pyroptosis-related proteins such as GSDMD and GSDME, as well as regulate the activation of both microglia and astrocytes (*Lyu, Bai & Qin, 2019*). Through these mechanisms, lncRNAs can initiate or amplify neuroinflammatory responses and pyroptotic neuronal death (*Zhang et al., 2021*). Therefore, investigating the interplay between lncRNAs and pyroptosis not only enhances understanding of PD pathophysiology but also presents novel opportunities for diagnostic and therapeutic intervention.

## ROLE OF ASTROCYTES AND OLIGODENDROCYTES IN PD

Beta-glucocerebrosidase (GBA1) is a membrane-bound lysosomal enzyme, and the accumulation of its substrates in neurons has been associated with cognitive impairments, motor and coordination deficits, and psychiatric disorders (*Orvisky et al., 2002*; *Schueler et al., 2003*). This accumulation triggers neurotoxicity, astrocytic hyperplasia, and neuroinflammation in both human patients and mouse models, with evidence of astrocyte proliferation and heightened inflammatory responses (*Farfel-Becker et al., 2014*, *2011*; *Wong et al., 2004*). Mutations in the GBA1 gene cause Gaucher's disease (GD), a lysosomal storage disorder, and represent one of the most common genetic risk factors for PD, increasing the likelihood of PD onset by 20- to 30-fold (*Stoker, Torsney & Barker, 2018*). Approximately 20% of adults with GD, including those with type 1, develop PD-related symptoms later in life (*Alaei et al., 2019*). Recent experimental studies provide compelling evidence that glial β-glucocerebrosidase plays a central role in both GD and GBA1-associated PD. Specifically, depletion of β-glucocerebrosidase in astrocytes and microglia leads to lysosomal dysfunction, neuroinflammation, α-syn accumulation, and impaired communication with neurons, all of which contribute to disease pathogenesis (*Aflaki et al., 2020*; *Boddupalli et al., 2022*; *Brunialti et al., 2021*; *Sanyal et al., 2020*; *Wang et al., 2022b*).

Oligodendrocytes, a type of glial cell, primarily function to envelop axons in the central nervous system (CNS) and form insulating myelin sheaths (*Duncan, Simkins & Emery, 2021*). Abnormalities in oligodendrocyte function can lead to neuronal damage (*Pandey et al., 2022*). Although clinical studies suggest a potential role for oligodendrocytes in neurodegenerative processes, research on their involvement in GBA1-related disorders remains relatively limited. Myelin, a lipid-rich insulating membrane, facilitates the saltatory conduction of action potentials and provides structural protection to axons (*Kuhn et al., 2019*). Due to the biochemical complexity of myelin, oligodendrocytes require
tightly regulated lipid synthesis and degradation pathways to ensure proper myelin formation and maintenance (*Meschkat et al., 2022*). In addition, they rely on specialized axon-glia communication pathways to deliver nutrients and organelles to distal regions of axons (*Edgar et al., 2021*).

Some studies have shown that oligodendrocyte impairments can lead to neurodegenerative features in mice, including axonal degeneration, microglial activation, astrocyte proliferation, and behavioral abnormalities. These findings suggest that oligodendrocytes play a significant role in neurodegeneration following the loss of β-glucocerebrosidase function, highlighting the critical contribution of oligodendrocyte dysfunction to the initiation and progression of GBA1-related neuropathology (*Zhang et al., 2024*).

PD has traditionally been considered a gray matter disorder; however, growing evidence supports the involvement of multiple glial cell types in the etiology of this synucleinopathy. Experimental studies have revealed significant effects on astrocytes, including impaired degradative capacity and dysfunctional inflammatory responses (*Aflaki et al., 2020*; *Sanyal et al., 2020*) In addition, microglia also exhibit compromised function in the context of PD, particularly due to defective β-glucocerebrosidase, which renders them unable to protect neurons from oxidative stress (*Brunialti et al., 2021*).

## ASTROCYTES REGULATE IRON DEATH, COPPER DEATH IN PD

Ferroptosis, a recently recognized form of regulated cell death, is characterized by iron accumulation and lipid peroxidation, and it plays a significant role in the degeneration of dopaminergic neurons in PD. In the autonomic nervous system, ferroptosis may also affect the vagus and parasympathetic nerves within specific tissues, potentially leading to functional impairments (*Zhang et al., 2022a*). In multiple PD models, treatment with ferrostatin-1 (Fer-1), a classical ferroptosis inhibitor, has been shown to reverse cell death (*Do Van et al., 2016*; *Mahoney-Sanchez et al., 2022*; *Sun et al., 2021*). Ferritin, the primary iron-sequestering protein, plays a pivotal role in maintaining iron homeostasis and is localized in various intracellular compartments or secreted extracellularly. Studies have shown that in primary-cultured astrocytes, iron exposure leads to increased intracellular iron levels, which triggers compensatory upregulation of ferritin secretion to buffer extracellular iron (*Wu et al., 2022*). In co-culture systems of primary astrocytes and MES 23.5 dopaminergic cells, ferritin released by astrocytes was found to enter the dopaminergic cells and exert a protective effect. These findings suggest that under iron overload, astrocyte-mediated ferritin secretion may mitigate oxidative stress and ferroptosis in dopaminergic neurons, a key process implicated in PD pathogenesis (*Zhang et al., 2022b*). NADPH oxidase 4 (NOX4), a prominent member of the NOX family, is highly expressed in astrocytes in the CNS (*Ma et al., 2017*). NOX4 is expressed predominantly in astrocytes (*Nayernia, Jaquet & Krause, 2014*). NOX4 activation contributes to ferroptosis by inducing mitochondrial dysfunction in astrocytes (*Wu et al.,*

2025), and elevated NOX4 levels can independently promote neuronal and astrocytic cell death (*Park et al., 2021*). Experimental evidence shows that NOX4 upregulation leads to mitochondrial dysfunction, initiating a cascade involving neuroinflammatory mediators such as osteopontin (OPN) and myeloperoxidase (MPO), which further promote astrocyte ferroptosis and accelerate PD progression (*Boonpraman et al., 2023*). With aging, astrocytes exhibit increased NOX activity and superoxide production, both of which contribute to PD pathogenesis (*Chekol Abebe et al., 2022*). Additionally, ionic imbalance in PD is thought to trigger ferroptotic cell damage in both neurons and oligodendrocytes. The presence of abnormal lipid-binding, disrupted redox balance, and elevated β-sheet-rich α-syn further supports the occurrence of ferroptosis in the PD microenvironment. These α-syn aggregates may interact with membrane lipids, compromise membrane integrity, and facilitate excessive $Ca^{2+}$ influx into mitochondria, thereby enhancing ROS production and triggering ferroptosis (*Fang et al., 2023*). This cascade of events amplifies inflammatory responses and immune dysregulation, further exacerbating the neurodegenerative progression of PD (*Zhang et al., 2024*).

Copper-mediated cell death, also known as cuproptosis, is a recently identified form of regulated cell death that bears mechanistic similarity to ferroptosis. Copper, an essential trace element, functions as a critical enzymatic cofactor in oxidative metabolism, antioxidant defense, and neurotransmitter biosynthesis. In cuproptosis, copper ions directly bind to thioctylated moieties on proteins involved in the TCA cycle, leading to the abnormal aggregation of these proteins and the concurrent depletion of iron–sulfur cluster-containing proteins. This interaction induces a proteotoxic stress response that ultimately results in cell death (*Tsvetkov et al., 2022*). Copper is vital for numerous neuronal processes, including modulation of synaptic proteins and neurotransmitter receptors (*Gaier, Eipper & Mains, 2013*). However, *in vitro* studies have shown that high concentrations of copper can promote the formation of partially folded, aggregation-prone amyloidogenic proteins (*Gromadzka et al., 2020*). Mechanistically, copper dyshomeostasis has been linked to toxic effects, particularly through the promotion of oxidative stress (*Giampietro et al., 2018*). As a result, precise regulation of copper levels is essential in preventing neurodegenerative processes associated with PD. This understanding has inspired novel therapeutic approaches aimed at restoring copper balance, such as the use of copper chelators, which have been shown to mitigate α-syn oligomer-induced ROS production (*Gromadzka et al., 2020*; *Deas et al., 2016*). In parallel, mitochondrial dysfunction—a central feature of PD pathophysiology—continues to be a major focus of therapeutic exploration, with several strategies aimed at enhancing the clearance of damaged mitochondria (*Malpartida et al., 2021*). For instance, PARK2 knockout in homozygous human induced pluripotent stem cells has been associated with disrupted TCA cycle activity, mitochondrial ultrastructural abnormalities, and increased oxidative stress (*Okarmus et al., 2021*). These findings support a close association between copper toxicity and PD, although the precise molecular mechanisms remain to be elucidated. Emerging research into genetic regulators, including key genes and microRNAs, may help

clarify the interplay between cuproptosis and PD. Given that both cuproptosis and ferroptosis involve mitochondrial dysfunction and excess acetyl-coenzyme A generation, it is plausible that these two pathways may be interrelated (*Zhang et al., 2022a*). Nevertheless, as both mechanisms are still in early stages of investigation, their specific contributions to PD pathology require further study.

## CLINICAL MANAGEMENT STRATEGIES IN PD

PD is predominantly characterized by the degeneration of dopaminergic (DA) neurons. One proposed therapeutic strategy involves replacing these lost neurons. A promising approach entails the use of microRNA (miRNA) to reprogram midbrain astrocytes into DA neurons (*Wei & Shetty, 2021*). Studies have demonstrated the efficacy of this method in both *in vivo* and *in vitro* models (*Ghasemi-Kasman et al., 2015*). However, despite its therapeutic potential, this strategy has several limitations. Notably, astrocyte reprogramming does not address the persistent presence of α-syn, which can propagate between cells and potentially infect newly generated neurons. Additionally, for this approach to be effective, the reprogrammed neurons must successfully integrate into existing neural circuits. Furthermore, *in vivo* reprogramming could lead to the depletion of astrocytes, which are essential for maintaining brain homeostasis, thereby limiting the overall therapeutic benefit of this strategy.

Currently, bioengineered exosomes are under investigation as a novel therapeutic approach for PD. Their ability to cross the BBB and their inherent biocompatibility make them suitable vectors for delivering therapeutic agents. This is particularly relevant considering the multifaceted roles of astrocytes, microglia, and α-syn in PD pathophysiology (*Wei & Shetty, 2021*). Bioengineered exosomes may modulate astrocytic inflammatory responses through various mechanisms. They have the capacity to regulate microglial polarization, prevent microglial overactivation, influence signal transduction between oligodendrocytes and neurons, and support effective nerve impulse conduction (*Afzal et al., 2025*). Through the regulation of these glial functions, bioengineered exosomes offer a promising avenue for modulating disease progression in PD.

In summary, this review emphasizes the central roles of astrocytes in both brain homeostasis and PD pathology (Fig. 4). Healthy astrocytes promote neuronal survival by releasing neurotrophic factors, clearing extracellular α-syn, regulating glutamate and fatty acid metabolism, and preserving BBB integrity. In contrast, dysfunctional astrocytes contribute to disease progression and neurodegeneration, while also affecting interactions with neurons and other glial cells. Notably, treatments such as deep brain stimulation may potentially mitigate ferroptosis and cuproptosis. Given the abundance of astrocytes in the brain, a comprehensive understanding of their complex and dynamic roles is essential for advancing therapeutic strategies against neurodegenerative disorders.

## FUTURE PROSPECTS AND LIMITATIONS

Although current research has illuminated some aspects of the interactions between astrocytes and other glial and neuronal cell types in PD, many mechanistic details remain elusive. Future studies should aim to elucidate the fine-tuned regulatory mechanisms

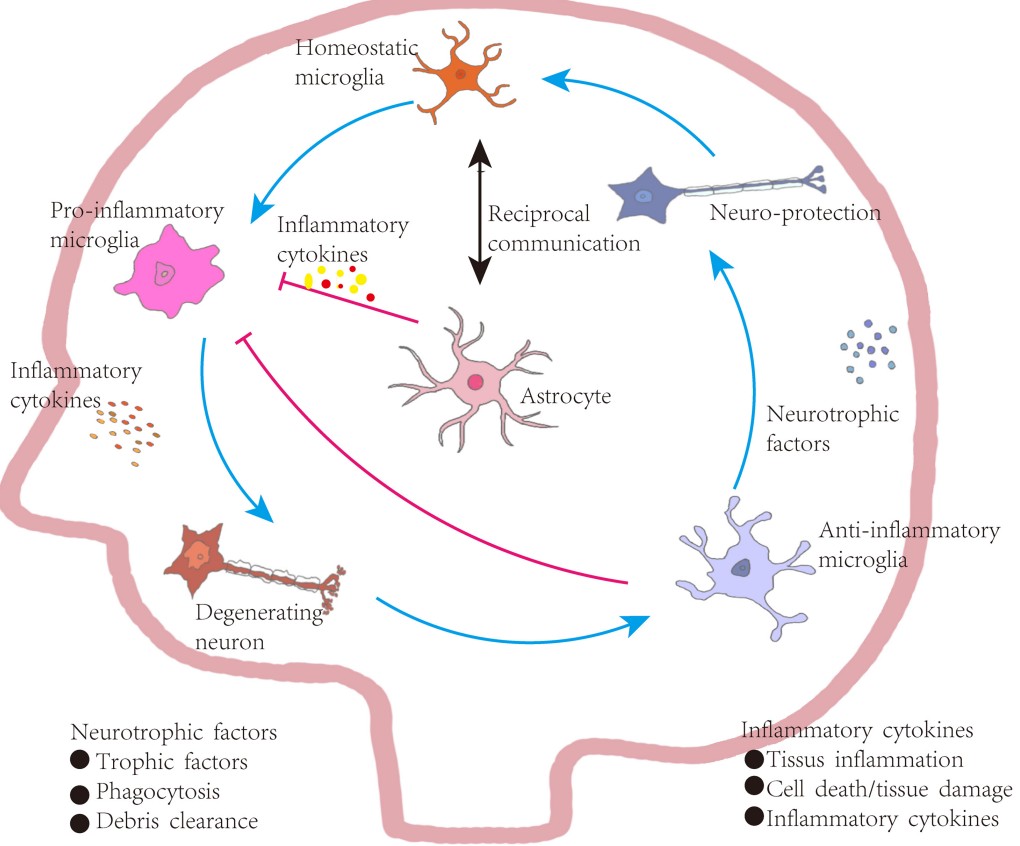

**Figure 4 Astrocyte-mediated regulation of inflammation in Parkinson's disease.** Under homeostatic conditions, microglia maintain a surveillant, non-inflammatory state. Upon exposure to pathological sti­muli, they transition into a pro-inflammatory phenotype, secreting cytokines that contribute to neuronal degeneration, tissue inflammation, and increased cellular apoptosis. In contrast, anti-inflammatory microglia support neuronal integrity by producing neurotrophic factors, engaging in phagocytosis, and clearing cellular debris. These functions aid in inflammation resolution, promote tissue repair, and restore microglial homeostasis, establishing a closed-loop regulatory system. Bidirectional communication exists between astrocytes and microglia. Astrocytes are responsive to cytokine signaling and can, in turn, regulate microglial activation. Some astrocyte subtypes release pro-inflammatory mediators that exacerbate neu­ronal injury. However, in PD, astrocytes are also implicated in suppressing excessive microglial activation through a negative-feedback mechanism. Activated microglia release high levels of pro-inflammatory cytokines and ROS, contributing to neuronal damage. *In vitro* studies indicate that astrocytes or astrocyte-conditioned media can attenuate microglial responses to inflammatory stimuli. Nevertheless, in PD and other neurodegenerative states, reactive microglia are abundant and produce cytokines such as interleukin-1α (IL-1α), which promote the transformation of astrocytes into neurotoxic, pro-inflammatory phenotypes.

underlying intercellular signaling pathways. For instance, it is crucial to determine how astrocyte-derived cytokines and chemokines modulate microglial activation states and functional responses. Furthermore, understanding the specificity of cell–cell interactions across different brain regions is essential for explaining region-specific pathological changes observed in PD. Such insights could enable the development of highly targeted modulators to precisely intervene in disease processes.

In parallel, ferroptosis and cuproptosis have emerged as significant contributors to PD pathology. However, the underlying mechanisms remain incompletely defined. Future investigations should focus on delineating the specific regulatory pathways governing ferroptosis and cuproptosis in various brain cell types—particularly dopaminergic neurons and astrocytes—as well as their relationships with other forms of cell death, such as apoptosis and pyroptosis. These insights may facilitate the design of novel therapeutic agents capable of precisely modulating these pathways, thereby opening new directions for PD treatment.

Another critical limitation in the current field is the lack of standardized research methodologies. Disparities in experimental protocols, such as differences in cell culture conditions, animal models, and evaluation metrics, hamper direct comparisons and integration of findings across studies. This lack of uniform standards reduces both the reproducibility and translational potential of research outcomes. Addressing this issue by establishing unified experimental frameworks will be instrumental in accelerating progress and ensuring the clinical relevance of future discoveries.

### Funding
This work has been funded by grants from the Central Government Guides Local Science and Technology Development funds (YDZX2022091). The funders had no role in study design, data collection and analysis, decision to publish, or preparation of the manuscript.

### Grant Disclosures
The following grant information was disclosed by the authors:
Central Government Guides Local Science and Technology Development: YDZX2022091.

### Competing Interests
The authors declare that they have no competing interests.

### Author Contributions
- Ziying Li conceived and designed the experiments, performed the experiments, prepared figures and/or tables, authored or reviewed drafts of the article, and approved the final draft.
- Mengran Cao performed the experiments, analyzed the data, authored or reviewed drafts of the article, and approved the final draft.
- Zhaoyang Yin analyzed the data, authored or reviewed drafts of the article, and approved the final draft.
- Xiaolei Li performed the experiments, prepared figures and/or tables, and approved the final draft.
- Qinglu Wang analyzed the data, authored or reviewed drafts of the article, and approved the final draft.
- Panpan Dong analyzed the data, authored or reviewed drafts of the article, and approved the final draft.

- Caixia Zhou conceived and designed the experiments, authored or reviewed drafts of the article, and approved the final draft.

## Data Availability

This is a literature review.

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
