# Peer review of "Functional characteristics, intercellular interactions and pathophysiological associations of astrocytes in Parkinson’s disease"

_PeerJ, doi:10.7717/peerj.19998_

## Round 0.1 · original submission · Major Revisions

**Language Note:** The review process has identified that the English language must be improved. PeerJ can provide language editing services - please contact us at [email protected] for pricing (be sure to provide your manuscript number and title). Alternatively, you should make your own arrangements to improve the language quality and provide details in your response letter. – PeerJ Staff

Reviewer 1 ·

Basic reporting

- The manuscript provides a comprehensive review on the role of astrocytes in Parkinson's disease (PD), addressing their functions, interactions, and pathological associations.

- A systematic literature search of recognized scientific databases (PubMed, Scopus, and Web of Science) has been performed, ensuring a comprehensive and well-supported coverage of the relevant literature.

- Multiple aspects of astrocytes are considered, including their role in the blood-brain barrier, regulation of α-synuclein, glutamate, and fatty acid metabolism, and their interaction with microglia.

Experimental design

- The document is well structured with clearly differentiated sections, which facilitates reading and understanding of the content.

- A solid theoretical structure is presented in the introduction, contextualizing the relevance of the topic and justifying the need for this review.

Validity of the findings

- It explores novel aspects such as the relationship of astrocytes with iron and copper death, which could open new lines of research in the treatment of Parkinson's disease.

- It addresses interactions between astrocytes and microglia that have not been systematically described in other recent reviews.

Additional comments

1. Linguistic and grammatical revision: A revision of the English language is recommended to improve the clarity and accuracy of the text. The assistance of a professional proofreader or a scientific editing service could be considered.

2. Include more figures and diagrams: Diagrams on the interactions between astrocytes, microglia, and neurons, as well as visual representations of the metabolic processes discussed, would help to improve the reader's comprehension.

3. Further development of the literature selection methodology: How the reviewed articles were selected should be described in greater detail, including inclusion/exclusion criteria and tools used to avoid bias.

4. Critical analysis of the literature: Incorporate a more in-depth discussion of controversies in the literature and the limitations of the reviewed studies to provide a more balanced and critical perspective.

5. Clinical applicability: Add a specific section on how these findings can be applied in the development of new therapies or strategies for the clinical management of Parkinson's disease.

6. Summarize in a final section the main conclusions and future directions: A closing section highlighting the key points reviewed and opportunities for future research would be beneficial to contextualize the relevance of the work.

Reviewer 2 ·

Basic reporting

This review attempts to describe the current state of understanding of the role of astrocytes in PD. Although this is an important topic, in its current state, there are many major flaws in the writing, listed below. The flaws are too numerous to give specific examples:
(1) Several sections are out of place / disjointed with the rest of the review.
(2) A lot of the writing is very choppy with abrupt, short sentences that also lack references.
(3) Many references have been completely missed or ignored
(4) There is a lot of irrelevant material on oligodendrocytes and microglia in PD, which is not the stated topic of this review.

Experimental design

The survey methodology is not sufficient since many of the important references to primary articles published in just the past 2 to 3 years have been completely missed.

Validity of the findings

-

Additional comments

In its current state, this review has numerous major flaws, and as such, does not rise to the standard of being able to give specific suggestions.

---

## Round 0.2 · Major Revisions

Reviewer 3 ·

Basic reporting

This is an excellent review that deserve to be published but the authors ignore important and relevant information about astrocytes neuroprotective role in Parkinson disease that was published in recent years, and that this review should include this review.
It has been suggested that astrocytes play a neuroprotective role in dopaminergic neurons that contain neuromelanin in the Parkinson's disease. The astrocytes secrete glutathione transferase M2-2 which together with DT-diaphorase prevents the neurotoxic effects of aminochrome in the dopaminergic neurons which contain neuromelanin, which are lost is in Parkinson’s disease. Aminochrome is a transient metabolite that is formed during the synthesis of neuromelanin in the dopaminergic neuron that can be neurotoxic by inducing mitochondrial dysfunction, formation of neurotoxic oligomers of alpha-synuclein, dysfunction of the protein degradation system both lysosomal and proteasomal systems, stress oxidative, neuroinflammation and endoplasmic reticulum. (see these publication;
doi:10.4103/1673-5374.335690. PMID: 35142659
doi:10.1007/s12640-020-00327-5. PMID: 33555546
doi:10.3390/antiox11020296. PMID: 35204179
doi:10.3390/antiox12030673. PMID: 36978921
doi:10.4103/1673-5374.380878. PMID: 37721280
doi:10.3390/biom14060673.PMID: 38927076
doi: 10.3390/antiox13091125.PMID: 39334784)

Experimental design

see my comments

Validity of the findings

see my comments

Additional comments

see my comments

---

## Round 0.3 · accepted · Accept

Both reviewers are completely satisfied by the revision. Revised manuscript is acceptable now.

Reviewer 1 ·

Basic reporting

Thank you to the authors for their replies.
The authors made significant changes to the manuscript, which improved it substantially.

Experimental design

The study design is well developed.

Validity of the findings

-It explores novel aspects such as the relationship of astrocytes with iron and copper death, which could open new lines of research in the treatment of Parkinson's disease.
Well developed

Reviewer 3 ·

Basic reporting

the manuscript has been improved and it is suitable for its publication

Experimental design

no applicable

Validity of the findings

This is an excelent review

Additional comments

The manuscript has been improved and it is suitable for its publication